# Diagnostic accuracy of GeneXpert MTB/RIF assay for detection of tubercular pleural effusion

**Jyoti Jain**[1☯]*, **Pooja Jadhao**[1☯], **Shashank Banait**[2‡], **Preetam Salunkhe**[1‡]

**1** Department of Medicine, Mahatma Gandhi Institute of Medical Sciences, Sewagram, Wardha, Maharashtra, India, **2** Department of Ophthalmology, Jawaharlal Nehru Medical College, Sawangi (Meghe), Wardha, Maharashtra, India

☯ These authors contributed equally to this work.
‡ These authors also contributed equally to this work.
* jyotijain@mgims.ac.in

## Abstract

India has been engaged in tuberculosis (TB) control activities for over 50 years and yet TB continues to remain India's important public health problem. The present study was conducted to compare the performance of GeneXpert MTB/RIF (GXpert) assay with composite reference standard in diagnosing cases of tubercular pleural effusion (TPE) and to evaluate the reliability of rifampicin resistance. A cross-sectional study was performed in a Department of Medicine of a rural teaching tertiary care hospital in central India. In all consecutive patients with pleural effusion on chest radiograph presenting to Department of Medicine, GXpert assay and composite reference standard was performed to evaluate the diagnostic accuracy of GXpert assay for detecting TPE in comparison to composite reference standard. Standard formulae were used to calculate the sensitivity, specificity, positive predictive values (PPV), negative predictive values (NPV), positive likelihood ratios (LR+) and negative likelihood ratios (LR-). Mc-Nemar's test was applied to compare variables. All comparisons were two-tailed. We considered the difference to be statistically significant if the P value was less than 0.05. The sensitivity of the GXpert assay in diagnosing TPE was 16.6% among 158 study participants, the specificity was 100% and diagnostic accuracy was 52.5% which was statistically significant (p value < 0.05). It had a PPV of 100% (95%CI: 88.3% - 100%) and a NPV of 47.5% (95%CI: 39.3% - 55.7%). The LR+ and LR-were 23.5 (95%CI: 1.43–38.6) and 0.83 (95%CI: 0.76–0.91) respectively. GXpert assay has a very high specificity in diagnosing TPE but has a low sensitivity. In comparison to composite reference standard Thus its clinical utility is limited when used as a standalone test. A physician's clinical acumen in combination with routine pleural fluid analysis should be the key factor in the diagnosis of TPE in clinically and radiologically suspected patients, especially in high TB burden countries.

**Data Availability Statement:** All relevant data are within the paper and its Supporting information files.

**Funding:** The authors received no specific funding for this work.

**Competing interests:** The authors have declared that no competing interests exist.

## Introduction

Tuberculosis is one of the oldest diseases known to affect humans [1]. India has been engaged in TB control activities for over 50 years and yet TB continues to remain India's important public health problem as it kills about 480,000 Indians every year.

Tubercular pleural effusion (TPE) is the second commonest form of extra pulmonary tuberculosis after TB lymphadenitis [2]. Major challenges in the diagnosis of extra pulmonary TB are its atypical clinical presentation, difficulties in obtaining specimens and the paucibacillary nature of the disease [3, 4]. Demonstrating the organism in pleural fluid by conventional microbiological techniques (staining and/or culture) have long been the cornerstone in diagnosing TPE [3]. These techniques have low sensitivity and long turnaround time of several weeks which is too long for a diagnostic test to be effective in curbing transmission [5]. In high TB burden settings, diagnosis of TPE is concluded by exudative pleural fluid with positive adenosine deaminase (ADA) levels along with lymphocytic predominance [4]. However, false-negative and false-positive results are a challenge.

As per Revised National Tuberculosis Control Programme (RNTCP) renamed as National TB Control Programme (NTEP) guidelines all presumptive extra-pulmonary TB in India microbiological confirmation should be done by CBNAAT, smear microscopy, culture and drug sensitivity testing for M tuberculosis and histopathological examination based on availability of specimen and facilities [6].

In early 2011, World health organization (WHO) endorsed a novel, rapid, automated, molecular diagnostic test, the GXpert assay, which can detect both *Mycobacterium tuberculosis* (MTB)complex and rifampicin resistance within 2 hours [7, 8]. Li et al. emphasized that compared with acid fast bacilli (AFB) smear and solid culture, GXpert assay has high sensitivities and short detection time, so could be used as an alternative for the rapid diagnosis of extra pulmonary TB in clinical practice [9]. However, Sehgal et al. [10] in a systematic review have strongly revoked the recommendations by WHO stating that neither the comparison of GXpert assay with culture as reference standard is justified (due to the low yield of culture in TPE) nor the comparison of GXpert assay with a composite reference standard is justified as GXpert assay has low sensitivity (22.7%).

The excellent diagnostic performance of GXpert assay has been demonstrated in sputum sample but not many studies have been undertaken for the evaluation of extra pulmonary samples and in the few studies that have been conducted in this context, there has been a wide variation in the findings in all these studies. In TB endemic countries like India, rapid case detection and decreasing the lag time in initiation of treatment is the key to TB Control. This is the first Indian study carried out in this area. The present study was conducted to compare the performance of GXpert assay with composite reference standard in diagnosing cases of TPE and to evaluate the reliability of rifampicin resistance.

## Materials and methods

### Ethics

The study was approved by the ethics committee of Mahatma Gandhi Institute of Medical Sciences (IRB00003623). We obtained a written informed consent from all study participants before enrolling them in the study.

### Setting

The study was conducted in the department of Medicine, Mahatma Gandhi Institute of Medical Sciences, Sevagram which is a 1000-bedded teaching tertiary care hospital located in a town

in central India. There are approximately 4, 50,000 outpatient visits and about 12,000 patients are admitted each year in department of Medicine in this Hospital.

## Study design

It was a cross-sectional observational, hospital based study which was carried out from October 1, 2016 to March 31, 2018.

## Study participants

All consecutive patients with pleural effusion on chest radiograph visiting Medicine OPD or patients admitted under Medicine Department were enrolled in present study. Patients with minimal pleural effusion which could not be tapped, history of bleeding diathesis or other contraindications to pleural fluid tapping and patients not willing to participate in the study were excluded.

## Sample size

Sample size was estimated using OpenEpi software based on following assumptions independently for sensitivity and specificity and the larger of the two was considered for the study purpose. Sample size was estimated to be around 158 as follows: hypothesized sensitivity of GeneXpert MTB/RIF assay for pleural TB = 0.60, hypothesized specificity of GeneXpert MTB/RIF assay for pleural TB = 0.99, absolute precision for both sensitivity and specificity = 0.25, set level of confidence = 95%, Z value associated with alpha = 1.96, prevalence of disease = 0.1and data loss while sample processing = 7% [11].

Study participants were recruited through consecutive sampling technique till the desired sample size of 158 was achieved. In all 176 patients with evidence of pleural effusion on chest radiograph had to be screened as per the inclusion criteria to get the desired 158 eligible participants. The reasons for not including 18 patients as per the exclusion criteria were: 8 patients with minimal pleural effusion which could not be tapped and 10 patients not willing to participate in the study = 10.

All study subjects with radiological evidence of pleural fluid were subjected to clinical, haematological and biochemical work up after detailed history and clinical examination. The management of the study subjects was carried out as per standard guidelines. We studied the complete blood count and other biochemical parameters like serum proteins and serum lactate dehydrogenase (LDH) required for Light criteria.

## Processing of pleural fluid samples

At the first visit of the study subjects, was tapped in all patients and the pleural fluid samples were subjected to index and reference tests simultaneously. Each sample was centrifuged and the pellet was aliquoted into parts as follows: 1 ml for GXpert assay, pleural fluid ADA, culture on Löwenstein–Jensen (LJ) medium, AFB on microscopy, pleural fluid LDH and pleural fluid protein for future reference. The initiation of anti-tubercular treatment (ATT) was not delayed pending results of GXpert assay. All patients were followed up at monthly intervals or on a required basis until completion of ATT.

## GXpert assay on pleural fluid samples (index test)

The GXpert assay was carried out using GeneXpert IV system (Cepheid Inc., Sunnyvale, CA, USA) as per the manufacturer's instructions. Briefly, to the 1 ml of sample, 2 ml of sample reagent was added. The mixture was vortexed for 30 seconds, then left to stand for 15 minutes

and finally added to the cartridge. The total turn-around time per sample was 1–2 hours and it was done free of cost. Standard myco-bacteriological procedures were followed the results to be estimated after which they were displayed on the computer screen and the results were distinguished among the following: no TB; TB detected, rifampicin resistance detected; TB detected, no rifampicin resistance detected; TB detected, rifampicin resistance indeterminate; and an invalid result.

## Composite reference standard on pleural fluid samples

Composite reference standard was used to evaluate the diagnostic accuracy of a GXpert assay in the absence of a perfect reference test. Composite reference standard included following tests: Exudative pleural fluid (by Lights criteria) with positive pleural fluid ADA level (Sensitivity 90–100%and a Specificity 89–100%) and/or positive for AFB on microscopy and/or positive culture on LJ medium and/or response to treatment. According to Light's criteria, the fluid is considered an exudate if: ratio of pleural fluid to serum protein greater than 0.5; ratio of pleural fluid to serum LDH greater than 0.6 or pleural fluid LDH greater than two thirds of the upper limits of normal serum value. The value of pleural fluid ADA > 42IU/L was considered as positive and was finalized according to our central laboratory.

Ziehl Neelsen (ZN) staining for smear microscopy of pleural fluid following the WHO recommended protocol was done. Patients with positive ZN smear of pleural fluid or growth of MTB on Mycobacterial Growth Indicator Tube (MGIT) liquid medium were also considered as cases of TPE.

All the study subjects considered as cases of TPE by exudative pleural fluid (by Lights criteria) with positive pleural fluid ADA level and/or Positive for AFB on microscopy were initiated on anti-tubercular treatment with four drugs- Isoniazid, Rifampicin, Pyrazinamide and Ethambutol for two months. All study subjects were followed up for 2 months after starting anti tubercular treatment and the response (decline in clinical symptoms and/or radiological clearing of pleural effusion) to treatment was seen at one month and two month. A final diagnosis was reached based on the composite reference standard and the study subjects were divided into two groups- TPE and non TPE with the following definitions: Tubercular pleural effusion was defined as an exudative pleural fluid (by Lights criteria) and positive pleural fluid ADA level and/or positive for AFB on microscopy along with response to anti—tubercular treatment. Pleural fluid analysis which did not meet the above criteria were considered as non—tubercular pleural effusion.

## Statistical analysis

We used SPSS software (version 16.0) to analyse the characteristics of the study population. Standard formulae were used to calculate the sensitivity, specificity, positive predictive values (PPV) and negative predictive values (NPV), positive likelihood ratios (LR+) and negative likelihood ratios (LR-). Mc-Nemar's test was applied to compare variables. All comparisons were two-tailed. We considered the difference to be statistically significant if the P value was less than 0.05.

## Results

We screened all consecutive patients with radiological evidence of pleural effusion during the study period and finally 158 of them were included in this study Fig 1. The baseline characteristics of the study subjects are summarized in Table 1. We observed that the mean age of the study subjects was 45 ± 17.3 years and that almost half of the study subjects were <40 years of age. There was male predominance with male to female ratio being 3.4:1. Majority (78.5%) of

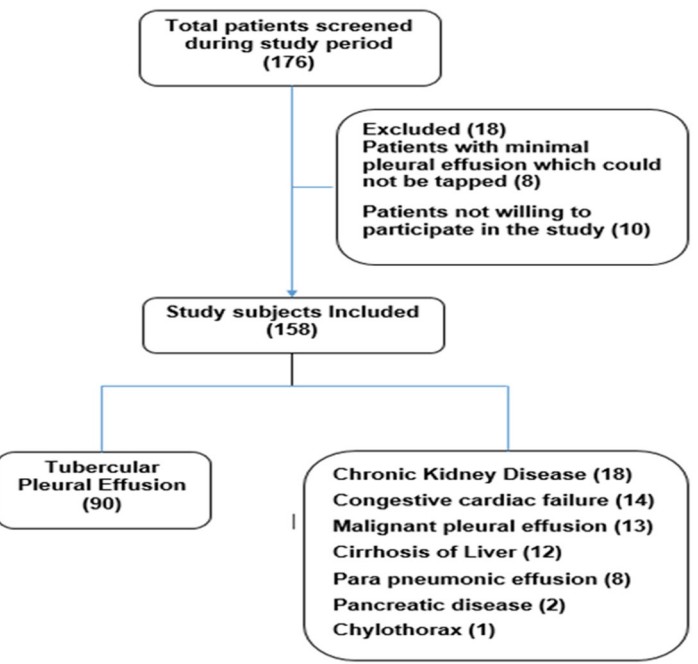

**Fig 1. Flow chart of the study subjects.**

study subjects had breathlessness as presenting symptom. Other presenting symptoms were cough of >15 day's duration and fever in 62% each, pleuritic chest pain in 35.4%, loss of appetite in 27.2% and night sweats present in 3.2%. None of the study subjects reported haemoptysis.

In our study we found that 70.3% (111/158) of the study subjects had exudative pleural effusion based on Light's criteria. Among these exudative pleural effusion 92.2% (83/111) study subjects had positive ADA level, and 81% (90/111) were responded to anti tubercular therapy. Our study revealed pleural fluid ZN microscopy for AFB as well as growth on MGIT liquid culture medium was negative in all the study subjects. Based on the composite reference standard 57% of study subjects had TPE, 11.4% had chronic kidney disease, 8.2% had malignant pleural effusion, 7.6% had cirrhosis of liver, 9% had congestive cardiac failure, 5% had para-pneumonic effusion, 1.3% had pancreatic effusion and 0.65% had chylothorax as shown in Table 1.

The GXpert assay gave a positive signal for the presence of MTB in 15/158 (9.5%) TPE study subjects. All these study subjects were also diagnosed to have TPE by composite reference standard as shown in Table 2.

The sensitivity and specificity of the diagnostic test GXpert assay was found to be 16.6% (95%CI: 8.9–24.3) and 100% (95%CI: 97.2–100) respectively in diagnosing TPE. It had a high PPV of 100% (95%CI: 88.3%–100%) and a NPV of 47.5% (95%CI: 39.3%–55.7%). The LR + and LR-were 23.5 (95%CI: 1.43–38.6) and 0.83 (95%CI: 0.76–0.91) respectively. The diagnostic accuracy of GXpert assay was found to be 52.5% which was statistically significant as evidenced by p value < 0.05. (Table 3) Of the 15 study subjects in which MTB was detected in pleural fluid, rifampicin resistance was detected by GXpert assay in 2 (13.3%) study subjects.

## Discussion

In our study we found the sensitivity of the GXpert assay in diagnosing TPE to be 16.6% and the specificity to be 100%. The low sensitivity and high specificity was supported by a P value

**Table 1. Demographical, biochemical, microbiological, clinical characteristics of study subjects along with GeneXpert MTB/RIF assay results.**

| Characteristic | Number (N = 158) | Percentage |
|---|---|---|
| **Age (completed years)** | | |
| 12–40 | 73 | 46.2 |
| 41–60 | 51 | 32.3 |
| 61 and above | 34 | 21.5 |
| **Sex** | | |
| Males | 122 | 78.2 |
| Females | 36 | 21.8 |
| **Symptoms of study subjects**[*] | | |
| Breathlessness | 124 | 78.5 |
| Cough | 98 | 62 |
| Fever | 98 | 62 |
| Pleuritic chest pain | 56 | 35.4 |
| Loss of appetite | 43 | 27.2 |
| Night sweats | 5 | 3.2 |
| **Nature of pleural fluid as per Light's criteria** | | |
| Exudate | 111 | 70.3 |
| Transudate | 47 | 29.7 |
| **Adenosine Deaminase level in pleural fluid** | | |
| Positive (>42.IU) | 83 | 52.5 |
| Negative (≥42.IU) | 75 | 47.5 |
| **Pleural fluid ZN microscopy for AFB** | | |
| AFB Positive | 0 | 0 |
| AFB Negative | 158 | 100 |
| **Pleural fluid culture Mycobacteria Growth Indicator Tube liquid medium** | | |
| MTB growth present | 0 | 0 |
| MTB growth absent | 158 | 100 |
| **Response to anti tubercular therapy (N = 93)** | | |
| Present | 90. | 96.8 |
| Absent | 3 | 3.2 |
| **Aetiology of pleural effusion determined by Composite Reference Standard** | | |
| Tubercular pleural effusion | 90 | 56.9 |
| Chronic kidney disease | 18 | 11.4 |
| Congestive cardiac failure | 14 | 8.9 |
| Malignant pleural effusion | 13 | 8.2 |
| Cirrhosis of liver | 12 | 7.6 |
| Syn-pneumonic effusion | 8 | 5.1 |
| Pancreatic effusion | 2 | 1.3 |
| Chylothorax | 1 | 0.6 |
| **GeneXpert MTB/RIF assay results in pleural fluid** | | |
| MTB detected | 15 | 9.5 |
| MTB not detected | 143 | 90.5 |
| **Result of Rifampicin resistance test in pleural fluid** | | |
| Rifampicin resistance detected | 2 | 13.3 |
| Rifampicin resistance not detected | 13 | 86.7 |

[*]Each patient may report one or more than one symptom.

**Table 2. Diagnosis of tubercular pleural effusion by GeneXpert MTB/RIF assay and composite reference standard in the study subjects (N = 158).**

| GeneXpert MTB/RIF | Composite reference standard | | Total | P value |
|---|---|---|---|---|
| | Positive | Negative | | |
| **Positive** | 15 | 0 | 15 | <**0.05** |
| **Negative** | 75 | 68 | 143 | |
| **Total** | 90 | 68 | 158 | |

of < 0.05. These values were similar to earlier reports which found a low sensitivity and high specificity of GXpert assay in detecting TPE as shown in Table 4 [10, 12–18]. The previous studies reported low sensitivity with a wide range from 15%–54.5% and all reported a very high specificity up to 100%. The low sensitivity of GXpert assay on pleural fluid could be explained by the low mycobacterial load leading to low DNA in pleural fluid. As expected the higher the mycobacterial load, it is more likely to obtain a positive result. Most of the studies conducted earlier reported a sensitivity similar to ours except for three studies. Rufai et al. found the sensitivity of GXpert assay to be 54.5% which is higher than our study [12]. Their study was conducted in a high TB burden area and included pleural fluid samples of patients who were TB suspects. Another study conducted by Du et al. found the sensitivity of GXpert assay to be 43.6% which is also relatively higher than in our study [14]. In addition to pleural fluid sample, they used pleural biopsy specimens as well which may have resulted in a relatively higher sensitivity. Trajman et al. in their study found the sensitivity of GXpert assay to be 3%, which very low as compared to our study [15]. The possible reason for this could be the presence of inhibitory substances in the pleural fluid as they did not submit the clinical specimens to any processing before running the test. Blood and other inhibitors, such as heparin or pus which are present in the collected sample could have interfered with cell lysis and cause inactivation of the DNA polymerase resulting in false-negative results. Also they did not use fresh pleural fluid sample and the long storage time in freezers could explain the low sensitivity.

Study demonstrates that a negative GXpert assay does not completely exclude the diagnosis of TPE as the test was unable to identify 83.4% (100–16.6) of confirmed TPE cases by composite reference standard (low sensitivity) and when GXpert assay was negative, 52.4% of patients still had TPE (low negative predictive value).

Pleural fluid GXpert assay to diagnose TPE has been evaluated in very few studies in India. The major strengths of our study are enrolment of all consecutive patients and all of them underwent same reference standard to which microbiologist and biochemist were blinded. We have followed up the patients to see the clinical response of anti-tubercular treatment only for

**Table 3. Diagnostic measures of GXpert assay in detecting tubercular pleural effusion as compared to composite reference standard (N = 158).**

| Measure | Value | 95% Confidence interval |
|---|---|---|
| **Sensitivity** | 16.6% | 8.9–24.3 |
| **Specificity** | 100% | 97.2–100 |
| **Positive Predictive Value** | 100% | 88.3–100 |
| **Negative Predictive Value** | 47.5% | 39.3–55.7 |
| **Positive Likelihood ratio** | 23.5 | 1.43–38.6 |
| **Negative likelihood ratio** | 0.83 | 0.76–0.91 |
| **Diagnostic accuracy** | 52.5% | |
| **P value** | < 0.05 | |

**Table 4. Comparison with previous studies conducted on the diagnostic utility of tuberculosis assay in detecting tubercular pleural effusion.**

| Sr. No | Author (Reference)/Year | Country | Studytype/ Samplesize | Meanage (± SD) years | Sexratio (M:F) | Sensitivity(%), Specificity(%) (95% CI) | PPV(%), NPV(%) (95% CI) | LR+, LR—(95%: CI) | Pvalue |
|---|---|---|---|---|---|---|---|---|---|
| 1 | Our Study/2018 | India | Cross-sectional /158 | 45±17.33 | 3.4:1 | 16.6(8.9–24.3), 100 (97.2–100) | 100(88.3–100), 52.4 (49.3–55.7) | 23.5(1.4–38.6) 0.83(0.76–0.91) | <0.05 |
| 2 | Rufai[12]/2015 | India | Prospective Observational/162 | 39±18 | 3:1 | 54.8 (38–70), 100 (85–100) | - | - | <0.01 |
| 3 | Sehgal[10]/2014 | India | Meta-analysis/24 | - | - | 22.7, 51.4 | - | - | - |
| 4 | Denkinger[18]/ 2013 | USA | Meta-analysis/841 | - | - | 21.4 (8–33), 98.7 (89–100) | - | - | - |
| 5 | Lusiba[15]/2013 | Uganda | Cross-sectional/116 | 34±13 | 1:1 | 28.7, 96.6 | 96.1,31.1 | - | <0.05 |
| 6 | Trajman[15]/ 2013 | Brazil | Cross-sectional /203 | 45±7.5 | 3:1 | 3 (0–17), 100 (89–100) | - | - | 0.02 |
| 7 | Porcel[17]/2012 | Spain | Case-Control/67 | 33 | 2:1 | 15 (7–32), 100 (88–100) | - | 11.3, 0.85 | 0.30 |
| 8 | Du [14]/2012 | China | Cross-sectional/134 | 38.6±12.3 | 1.25:1 | 43.6,98.6% | 96,69.3 | - | <0.01 |
| 9 | Friedrich[13]/ 2011 | South Africa | Cross-sectional /25 | - | - | 25,100 | - | - | <0.001 |
| 10 | Meldau [16]/2012 | South Africa | Prospective Cohort/ 103 | 39±10 | 3:2 | 22.5 (12.4–37), 98 (89–99.7) | 91.4, 69.7 | - | <0.05 |

two months and thus confirmed our diagnosis which has not been done in any of the previous studies. Also the sample collection was done under stringent measures and immediate transportation to the laboratory without any time lag was done, improving the chances of a positive test as a time delay in initiating the test or use of frozen pleural fluid sample could lead to false negative results.

The study had a few limitations. To reach a final diagnosis of TPE we used a composite reference standard as opposed to the gold standard which includes histopathological demonstration of AFB in pleural tissue or a growth of MTB on culture medium. To conduct a pleural biopsy in all patients was not feasible in our setting and although we did use pleural fluid culture, its results were limited due to a poor diagnostic yield. We did not perform some of the tests that are known to be indicative of TPE including markers such as Interferon gamma assays because these are not readily available in our setting. The results of our study cannot be generalized as it was conducted in one center.

Based on our study results we cannot recommend GXpert assay for diagnosis of TPN but can be used to rule out TPN in resource constrained settings as it has high specificity. Research on more cost-effective methods singly or in combination for early detection of TPN in patients with pleural effusion should be conducted. A multicentric study with larger sample size including patients varying TB burden could help achieve a more accurate result. Also further evaluation of diagnostic algorithms in different epidemiological and geographical settings and patient populations can be undertaken. Its cost-effectiveness and cost-benefit analyses along with its impact on the diagnosis of TB, multidrug resistant TB and the management of patients can be studied.

## Conclusion

In the light of our findings we have reached to the conclusion that GXpert assay has a very high specificity in diagnosing TPE but has a low sensitivity. Thus its clinical utility is limited when used as a standalone test. Rifampicin resistance in pleural fluid was detected by GXpert assay but not by routine drug sensitivity testing. A physician's clinical acumen in combination

with routine pleural fluid analysis should be the key factor in the diagnosis of TPE in clinically and radiologically suspected patients, especially in high TB burden countries.

## Supporting information

**S1 Dataset.**
(XLSX)

## Acknowledgments

We wish to thank our patients for their cooperation in this study.

## Author Contributions

**Conceptualization:** Jyoti Jain, Pooja Jadhao.

**Data curation:** Jyoti Jain, Pooja Jadhao, Preetam Salunkhe.

**Formal analysis:** Jyoti Jain.

**Investigation:** Jyoti Jain, Pooja Jadhao.

**Methodology:** Jyoti Jain, Pooja Jadhao, Preetam Salunkhe.

**Project administration:** Jyoti Jain, Pooja Jadhao.

**Resources:** Jyoti Jain.

**Software:** Jyoti Jain, Shashank Banait, Preetam Salunkhe.

**Supervision:** Jyoti Jain, Shashank Banait, Preetam Salunkhe.

**Validation:** Jyoti Jain, Shashank Banait, Preetam Salunkhe.

**Visualization:** Jyoti Jain, Shashank Banait.

**Writing – original draft:** Jyoti Jain, Preetam Salunkhe.

**Writing – review & editing:** Jyoti Jain, Pooja Jadhao, Shashank Banait, Preetam Salunkhe.

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
