## [Decision Letter · Decision Letter 0]

28 Jan 2020

PONE-D-19-33113

Diagnostic accuracy of GeneXpert MTB/RIF assay for detection of tubercular pleural effusion

PLOS ONE

Dear Dr Jain,

Thank you for submitting your manuscript to PLOS ONE. After careful consideration, we feel that it has merit but does not fully meet PLOS ONE’s publication criteria as it currently stands. Therefore, we invite you to submit a revised version of the manuscript that addresses the points raised during the review process.

You have been provided comments by both the reviewers. In addition, i have shared the submitted pdf with my comments using sticky notes. 

We would appreciate receiving your revised manuscript by 17 Feb 2020. To enhance the reproducibility of your results, we recommend that if applicable you deposit your laboratory protocols in protocols.io, where a protocol can be assigned its own identifier (DOI) such that it can be cited independently in the future. For instructions see: http://journals.plos.org/plosone/s/submission-guidelines#loc-laboratory-protocols

We look forward to receiving your revised manuscript.

Kind regards,

Hemant Deepak Shewade, MBBS MD

Academic Editor

PLOS ONE

Additional Editor Comments:

I have shared the submitted pdf with my comments using sticky notes. 

2. We noticed you have some minor occurrence(s) of overlapping text with the following previous publication(s), which needs to be addressed:

https://doi.org/10.1016/j.tube.2018.09.002

In your revision ensure you cite all your sources (including your own works), and quote or rephrase any duplicated text outside the Methods section. Further consideration is dependent on these concerns being addressed.

4. Please upload a copy of Supporting Information S1 Figure which you refer to in your text on page 15.

Reviewers' comments:

Reviewer's Responses to Questions

**Comments to the Author**

1. Is the manuscript technically sound, and do the data support the conclusions?

Reviewer #1: Partly

Reviewer #2: Yes

2. Has the statistical analysis been performed appropriately and rigorously? 

Reviewer #1: No

Reviewer #2: No

3. Have the authors made all data underlying the findings in their manuscript fully available?

Reviewer #1: No

Reviewer #2: Yes

4. Is the manuscript presented in an intelligible fashion and written in standard English?

Reviewer #1: Yes

Reviewer #2: Yes

5. Review Comments to the Author

Reviewer #1: The manuscript describes an important subject relevant to public health importance. However, the manuscript needs some editing for clarity. The authors may consider following comments for their manuscript:

Major comments:

- Introduction needs restructuring. The reference to India has been provided in beginning of the introduction, however, no further mention of national TB programme of country or relevance to country is given. The authors can strengthen the rationale of the study, if it is aimed towards India or for global evidence.

- The recruitment process must be explained in the manuscript. If sample size was calculated as 158, how did the researchers stop the recruitment once it reached the sample size...

- Discussion: The authors must provide key policy and practice recommendations for national programme or at global level, based on the results.

Minor comments:

- Reference style and reference lists must be updated. For example, Reference 2, 3 does not reflect the statement in manuscript. References do not have complete title of the manuscripts. Please review

- Abbreviations must be mentioned first along with the full form in parenthesis. Line 46 mentioned TPE, however it must be mentioned in Line 42 first.

- Line 78-79 is a repeat of Line 73-75. Please review

- Line 79-81 could be mentioned under Study participants, and not Study design

- Line 117-118 needs a reference.

- The title of figures and tables must be stand-alone. It must include information about the study site and time period.

- Table 1:

o There is an overlap in age categories: It must be 61 and above and not 60 and above

o Symptoms of study subjects: It is understood that each patient may report one or more than one symptom. This must be mentioned in the footnote of the table.

- Line numbers are not continuous, the line numbers have restarted from Page 13. Please review

- Page 14, Line 25-26 are repeat of Page 13 Line 4-5. Please edit.

- Page 13, Line 7: Not sure if the use of standardized protocol and adequate sample size is a major strength of the study. It is part of methodology and a study must follow the protocol.

- The authors may add that the study is not generalizable to other contexts under Limitations, considering this study was performed in one center of the country.

END

Reviewer #2: The article is important because it provides evidence on the sensitivity of Xpert MTB/RIF assay among Extrapulmonary TB (EPTB). The End TB strategy targets all types of TB and not only bacteriologically confirmed, therefore knowing the sensitivity and predictive values of Xpert MTB/RIF is key to reach the true patients suffering from (EPTB) and allowing reaching targets of End TB strategy. Also, the study is performed in a high TB burden country with limited resources. Those characteristics are common in high burden countries. In that sense, the study suggests a composite score for diagnosing EPTB that could be very useful in similar contexts.

There is room to improve the quality of the article. I suggest to better explain the calculation of sample size. Were the two groups, one Xpert MTB/RIF were compared to composite? if yes, what was the sensitivity assumed in each group?

In the results part, the negative predictive value (NPV) the figure should be revised. If there were 68 true negatives divided by true negatives plus false negatives (75), NVP should be 47.55% and not 52.4%. If the numbers are provided aside to the percentage this could improve the understanding of the calculation. Then, the likelihood (Sensitivity/1-Specificity) positive ratio would be -0.83 and not 23.5. These issues need to be revised.

Concerning the clinical response, it would be useful to have the treatment outcomes at the end of full treatment completion.

In conclusions: nothing is mentioned about rifampicin resistant diagnosed patients, which was one of the objectives of the study.

Some format comments:

Line 3: Capital in ‘jain’ = Jain

Line 11: no need to repeat ‘& These authors…” because it seems that all authors contributed equally.

Line 58: superscript ‘11’ not clear if it is reference as the references are formatted in brackets.

In tables: there are full stops in name of categories that should be removed.

Reference 12 is cited as WHO when the source comes from J Med Microbiol.

6. PLOS authors have the option to publish the peer review history of their article (what does this mean?). If published, this will include your full peer review and any attached files.

Reviewer #1: No

Reviewer #2: Yes: Nimer Ortuño-Gutiérrez

---

## [Author Response · Author response to Decision Letter 0]

1 Jul 2020

The authors would like to thank the academic editor and reviewers for their specific and helpful comments. 

Manuscript no: PONE-D-19-33113

The manuscript has been improved according to the suggestions:

Editor comments-

1) We have ensured that manuscript meets PLOS ONE’s style requirements and have made the necessary corrections to our manuscript.

2) Regarding the occurrence of overlapping text with previous publication we have rephrased the sentence and cited the original source.

3) As per your suggestion we will change to our Data Availability statement, in our cover letter. 

4) Supporting information S1 figure has been uploaded after edited through Preflight Analysis and Conversion Engine (PACE) digital diagnostic tool. 

5) We restructured the introduction as per your suggestion . and mentioned the national TB programme of country .

6) We have explained the recruitment process and sample size calculation in detail.

7) In the Discussion section we have provided key policy and practice recommendations for national programme based on the results of our study.

8) Line 61- We have clarified the need for this study by adding the following points- ‘although few studies have been conducted in this area, there has been a wide variation in the findings in all the studies’ 

Minor comments:

1. We introduced all abbreviations when mentioned first along with the full form in parenthesis. 

2. Line 78-79 is a repeat of Line 73-75, so it was deleted. 

3. Line 79-81 changed as part of study participants. 

4. Line 117- 118 we have made the correction and the value of pleural fluid ADA > 42IU/L was finalized according to our central laboratory. 

5. Page 14, Line 25-26 are deleted

6. Page 13, Line 7: Removed as a major strength of the study as it is part of methodology.

We have mentioned that the results of our study is not generalizable to other contexts under Limitations, considering this study was performed in one center of the country.

Reviewer #1: 

Major comments:

1) We have restructured our introduction as per your suggestion.

2) The recruitment process is explained in the revised manuscript along with the detailed method for sample size calculation.

3) We have provided key policy and practice recommendations in discussion part as suggested.

1) Minor comments:

 Reference style and reference lists has been updated. 

2) We have mentioned abbreviations in parenthesis at the first mention, have made the 

3) necessary corrections in tables and modified the strenghts of the study. 

4) Table 1: As suggested by the reviewer, we changed the overlap in age categories: from 60 

5) and above to 61 and above.

6) The title has been changed as per your suggestions for Table 1. We have also added footnote.

7) We made the changes in line numbers, now they are continuous. 

Reviewer #2:

Table 1 –Addition of 2x2 contingency table for better clarity as suggested has been added.

GeneXpert MTB/RIF Composite reference standard Total P value

 Positive Negative 

Positive 15 0 15 <0.05

Negative 75 68 143 

Total 90 68 158 

1) Line 182-189 and table 3: we have incorporated gold standard of other studies advised.

2) Line 8-9, page 13- Of the total 158 study subjects 93 were started on anti-tubercular therapy based on clinical findings, exudative pleural effusion by Light’s criteria and positive pleural fluid ADA level. Of the 93 subjects, 90 (96.7%) responded to anti tubercular treatment while remaining 3 subjects did not respond.

3) Line 24, page 14- the correction suggested has been incorporated in our manuscript.

4) Reference style has been changed according to PLOS ONE guidelines.

Reviewer 2 comments- 

1) Detailed method for sample size calculation along with sensitivity assumed in each group is mentioned in revised manuscript.we have compared Xpert MTB/RIF to composite reference to calculate various parameters of diagnostic study.

2) We have corrected the error in negative predictive value and positive likelihood ratio as suggested by you.

3) In conclusions section we have mentioned about rifampicin resistant as suggested by you.

4) We have mentioned that first two authors have contributed equally and the third and fourth authors have contributed equally to the work done.

5) We introduced all abbreviations, reference and corrected grammatical mistakes

We hope that our modifications render our manuscript in its current form suitable for publication in PLOS ONE.

Yours sincerely,

Dr Jyoti Jain

---

## [Editor Report · Decision Letter 1]

5 Aug 2020

PONE-D-19-33113R1

Diagnostic accuracy of GeneXpert MTB/RIF assay for detection of tubercular pleural effusion

PLOS ONE

Dear Dr. Jain,

Kindly resubmit.

The response to review comments file is not formatted well enough to differentiate between reviewer comment and author resposne.

I suggest this

Please repond point by point. First mention the reviewer comment. Then mention your author response below. Then mention the second comment by reviewer and then provide provide author response to the second comment.

Mention in capital REVIEWER COMMENT before a reviewer comment and then follow it up with AUTHOR RESPONSE in capital before providing a response.

Give a line space between one set of REVIEWER COMMENT and AUTHOR RESPONSE.

Do this uniformly throughout the response to reviewer comment file.

In the author response, please mention the exact line where the edits have been made in the revised mansucript with track changes.

Kind regards,

Hemant Deepak Shewade, MBBS MD

Academic Editor

PLOS ONE

---

## [Author Response · Author response to Decision Letter 1]

14 Aug 2020

14th of August 2020

The authors would like to thank the academic editor and reviewers for their specific and helpful comments. 

Please find enclosed the edited manuscript in word format with track changes.

Title: Diagnostic accuracy of GeneXpert MTB/RIF assay for detection of tubercular pleural

Effusion.

Author: Jyoti* Jain, Pooja$ Jadhao, Shashank^ Sharadchandra Banait, Preetam# Salunkhe.

Manuscript no: PONE-D-19-33113

The manuscript has been improved according to the suggestions:

ACADEMIC EDITORS ADDITIONAL COMMENTS

Editor Comments 1: line 61 - needs a review. Check for studies on Xpert and EP samples. 

Please look at your table 3. There are studies. Please provide a summary of the table here and mention what is 'new' in your study? 'Why' this study? Purpose of the study

AUTHOR RESPONSE: Line 61- We have clarified the need for this study by adding the following points- ‘although few studies have been conducted in this area, there has been a wide variation in the findings in all the studies’. Page 4, Line 64-68.

Editor Comments 2: Line 73-75, not part of settings.

AUTHOR RESPONSE: Line 73-75 are removed from setting.

Editor Comments 3: Line 92- check the sentence

AUTHOR RESPONSE: Line 92 sentence restructured. Page 6, Line 106.

Editor Comments 4: Line 117 ref

AUTHOR RESPONSE: Line 117- 118 we have made the correction and the value of pleural fluid ADA > 42IU/L was finalized according to our central laboratory. Page 7, Line 131-132.

Editor Comments 5: line 119-120 ref

AUTHOR RESPONSE: Reference added and sentence restructured. Krishna M, Gole SG. Comparison of conventional Ziehl–Neelsen method of acid fast bacilli with modified bleach method in tuberculous lymphadenitis. Journal of cytology. 2017 Oct;34(4):188.

Editor Comments 6: Tables and figure titles, Please ensure that the titles are standalone. Ensure a time, place person component in title. Add appropriate footnotes.

AUTHOR RESPONSE: The title has been changed as per your suggestions for Tables and figures. Appropriate footnotes added.

Editor Comments 7: Table 1. Use meaningful age cut offs

say <10

10-19 

20 and above

OR

<15

15-44

45-64

65 and above

AUTHOR RESPONSE: In our hospital setting age group upto 12 years are admitted in paediatrics department and in the department of medicine we cater patients with ≥ 12 years of age. As this study included patients from department of medicine we have kept these age cut offs.

Editor Comments 8: In addition to the table 2 presented by the authors, 

Show the standard two by two table as a separate table. 

Column heading - TPE+ TPE-

Row heading - Xpert+ XPert-

AUTHOR RESPONSE: 2x2 contingency table for better clarity as suggested has been added. Page 11, Line 179-180.

GeneXpert MTB/RIF Composite reference standard Total P value

 TPE Positive TPE Negative 

Positive 15 0 15 <0.05

Negative 75 68 143 

Total 90 68 158 

Editor Comments 9: Authors used a composite index for TPE and used it as a gold standard. 

What was the gold standard used in these studies?

AUTHOR RESPONSE: We have incorporated gold standard of other studies as advised in Table 3.

Editor Comments 10: Table 3. Please also add a column heading and mention what gold standard was used in the respective studies?

AUTHOR RESPONSE: In table 3, we have incorporated gold standard of other studies as advised.

Editor Comments 11: Line 8-9, page 13.almost all with TPE+ (based on composite index) responded to ATT. This line has to be specifically there in results narrative (give numerator and denominator)

AUTHOR RESPONSE: Of the total 158 study subjects 93 were started on anti-tubercular therapy based on clinical findings, exudative pleural effusion by Light’s criteria and positive pleural fluid ADA level. Of the 93 subjects, 90 (96.7%) responded to anti tubercular treatment while remaining 3 subjects did not respond. Page 4, Line 64-68.

Editor Comments 12: Line 10, page 13

low time lag = is this supported by data?

AUTHOR RESPONSE: Yes, this is supported by study done by Friedrich et al published in J Clin Microbiol in 2011. According to this study statistically significant difference (P= 0.0021, paired two tailed Student’s t test)is found due to storage of sample and frozen samples showed false negative results for M tuberculosis by Xpert assay. We are adding reference to this sentence also. Page 16, Line 222.

Editor Comments 13: Line 24, page 14

Limited clinical utility as a standalone test expecially if the Xpert results are negative (specify this)

AUTHOR RESPONSE: Line 24, page 14- the correction suggested has been incorporated in our manuscript.

REVIEWER #1: MAJOR COMMENTS

REVIEWER COMMENT1: Introduction needs restructuring. The reference to India has been provided in beginning of the introduction, however, no further mention of national TB programme of country or relevance to country is given. The authors can strengthen the rationale of the study, if it is aimed towards India or for global evidence.

AUTHOR RESPONSE: We have restructured our introduction as per your suggestion and mentioned the national TB programme of country. Page 3, Line 49-53, Page 4, Line 64-68. 

REVIEWER COMMENT 2: The recruitment process must be explained in the manuscript. If sample size was calculated as 158, how did the researchers stop the recruitment once it reached the sample size.

AUTHOR RESPONSE: The recruitment process is explained in the revised manuscript along with the detailed method for sample size calculation. Page 5, Line 88-100.

REVIEWER COMMENT3: Discussion: The authors must provide key policy and practice recommendations for national programme or at global level, based on the results.

AUTHOR RESPONSE: We have provided key policy and practice recommendations in discussion part as suggested. Page 16, Line 230-232.

REVIEWER #1: MINOR COMMENTS

REVIEWER COMMENT 1: Reference style and reference lists must be updated. For example, Reference 2, 3 does not reflect the statement in manuscript. References do not have complete title of the manuscripts. Please review

AUTHOR RESPONSE: Reference style and reference lists has been updated. Page 17-18, Line 249-293.

REVIEWER COMMENT 2: Abbreviations must be mentioned first along with the full form in parenthesis. Line 46 mentioned TPE, however it must be mentioned in Line 42 first.

AUTHOR RESPONSE: We introduced all abbreviations when mentioned first along with the full form in parenthesis. Page 3, Line 39.

REVIEWER COMMENT 3: Line 78-79 is a repeat of Line 73-75. Please review

AUTHOR RESPONSE: Line 78-79 is a repeat of Line 73-75, so it was deleted. 

REVIEWER COMMENT 4: Line 79-81 could be mentioned under Study participants, and not Study design.

AUTHOR RESPONSE: Line 79-81 deleted from study design. 

REVIEWER COMMENT 5: Line 117-118 needs a reference.

AUTHOR RESPONSE: Line 117- 118 we have made the correction and the value of pleural fluid ADA > 42IU/L was finalized according to our central laboratory. Page 7, Line 132-133.

REVIEWER COMMENT 6: The title of figures and tables must be stand-alone. It must include information about the study site and time period.

AUTHOR RESPONSE: The title has been changed as per your suggestions for Tables and figures.

REVIEWER COMMENT 7: Table 1: o There is an overlap in age categories: It must be 61 and above and not 60 and above. o Symptoms of study subjects: It is understood that each patient may report one or more than one symptom. This must be mentioned in the footnote of the table.

AUTHOR RESPONSE: Table 1: As suggested by the reviewer, we changed the overlap in age categories: from 60 and above to 61 and above. We have also added footnote to Table 1 as per your suggestion. Page 9, Table 1.

REVIEWER COMMENT 8: Line numbers are not continuous, the line numbers have restarted from Page 13. Please review

AUTHOR RESPONSE: We made the changes in line numbers, now they are continuous. 

REVIEWER COMMENT 9: Page 14, Line 25-26 are repeat of Page 13 Line 4-5. Please edit.

AUTHOR RESPONSE: Page 14, Line 25-26 is a repeat of Page 13 Line 4-5, so it was deleted.

REVIEWER COMMENT 10: Page 13, Line 7: Not sure if the use of standardized protocol and adequate sample size is a major strength of the study. It is part of methodology and a study must follow the protocol.

AUTHOR RESPONSE: Page 13, Line 7: Removed as a major strength of the study as it is part of methodology.

REVIEWER COMMENT 11: The authors may add that the study is not generalizable to other contexts under Limitations, considering this study was performed in one center of the country.

AUTHOR RESPONSE: We have mentioned that the results of our study is not generalizable to other contexts under Limitations, considering this study was performed in one center of the country. Page 16, Line 228-229.

REVIEWER #2: MAJOR COMMENTS

REVIEWER COMMENT 1: The article is important because it provides evidence on the sensitivity of Xpert MTB/RIF assay among Extrapulmonary TB (EPTB). The End TB strategy targets all types of TB and not only bacteriologically confirmed, therefore knowing the sensitivity and predictive values of Xpert MTB/RIF is key to reach the true patients suffering from (EPTB) and allowing reaching targets of End TB strategy. Also, the study is performed in a high TB burden country with limited resources. Those characteristics are common in high burden countries. In that sense, the study suggests a composite score for diagnosing EPTB that could be very useful in similar contexts.

AUTHOR RESPONSE: Detailed method for sample size calculation along with sensitivity assumed in each group is mentioned in revised manuscript. Page 5, Line 88-100.

REVIEWER COMMENT 2: There is room to improve the quality of the article. I suggest to better explain the calculation of sample size. Were the two groups, one Xpert MTB/RIF were compared to composite? if yes, what was the sensitivity assumed in each group?

AUTHOR RESPONSE: Detailed method for sample size calculation is mentioned in revised manuscript as per reviewer suggestion. We have compared Xpert MTB/RIF to composite reference to calculate various parameters of diagnostic study. Page 5, Line 88-100.

REVIEWER COMMENT 3: In the results part, the negative predictive value (NPV) the figure should be revised. If there were 68 true negatives divided by true negatives plus false negatives (75), NVP should be 47.55% and not 52.4%. If the numbers are provided aside to the percentage this could improve the understanding of the calculation. Then, the likelihood (Sensitivity/1-Specificity) positive ratio would be -0.83 and not 23.5. These issues need to be revised.

AUTHOR RESPONSE: We have corrected the error in negative predictive value and positive likelihood ratio as suggested by you. Page 13, Table 3 Row 5 and 6.

REVIEWER COMMENT 4: Concerning the clinical response, it would be useful to have the treatment outcomes at the end of full treatment completion.

AUTHOR RESPONSE: I also agree with your opinion regarding clinical response to the treatment at the end of full treatment completion. However being a cross sectional observational study, we did not followed up the study participants for 6months however among patients in whom Anti tubercular treatment was started they were followed up to 2 months of intensive phase of treatment. Page 16, Line 219.

REVIEWER COMMENT 5: In conclusions: nothing is mentioned about rifampicin resistant diagnosed patients, which was one of the objectives of the study.

AUTHOR RESPONSE: In conclusions section we have mentioned about rifampicin resistant as suggested by you. Page 17, Line 240.

REVIEWER #2: FORMAT COMMENTS

REVIEWER COMMENT 1: Line 3: Capital in ‘jain’ = Jain

AUTHOR RESPONSE: Changes done as per suggestion. Page 1, Line 3.

REVIEWER COMMENT 2: Line 11: no need to repeat ‘& These authors…” because it seems that all authors contributed equally.

AUTHOR RESPONSE: ¶These authors contributed equally to this work (means Jyoti Jain and Dr Pooja Jadho)

& These authors also contributed equally to this work. (means Shashank Banait and Preetam salunkhe)

REVIEWER COMMENT 3: Line 58: superscript ‘11’ not clear if it is reference as the references are formatted in brackets.

AUTHOR RESPONSE: Changes done as per suggestion and reference is formatted as per PLoS guidelines.

REVIEWER COMMENT 4: In tables: there are full stops in name of categories that should be removed.

AUTHOR RESPONSE: In tables full stops in name of categories are removed as suggested by reviewer.

REVIEWER COMMENT 5: Reference 12 is cited as WHO when the source comes from J Med Microbiol.

AUTHOR RESPONSE: Changes done as suggested by reviewer.

We hope that our modifications render our manuscript in its current form suitable for publication in PLOS ONE.

Yours sincerely,

Dr Jyoti Jain

---

## [Decision Letter · Decision Letter 2]

2 Oct 2020

PONE-D-19-33113R2

Diagnostic accuracy of GeneXpert MTB/RIF assay for detection of tubercular pleural effusion

PLOS ONE

Dear Dr. Jain,

Thank you for submitting your manuscript to PLOS ONE. After careful consideration, we feel that it has merit but does not fully meet PLOS ONE’s publication criteria as it currently stands (Minor Revision). Therefore, we invite you to submit a revised version of the manuscript that addresses the points raised during the review process.

We look forward to receiving your revised manuscript.

Kind regards,

Hemant Deepak Shewade, MBBS MD

Academic Editor

PLOS ONE

Additional Editor Comments (if provided):

Please address reviewer 1 comments

The abstract has to be formatted as per plos one guidelines

The revised abstract should incorporate the edtis made in the main text of the manuscript. Please mention the number of study population in the abstract results. Currently this is missing.

Reviewers' comments:

Reviewer's Responses to Questions

**Comments to the Author**

1. If the authors have adequately addressed your comments raised in a previous round of review and you feel that this manuscript is now acceptable for publication, you may indicate that here to bypass the “Comments to the Author” section, enter your conflict of interest statement in the “Confidential to Editor” section, and submit your "Accept" recommendation.

Reviewer #1: (No Response)

2. Is the manuscript technically sound, and do the data support the conclusions?

Reviewer #1: Yes

3. Has the statistical analysis been performed appropriately and rigorously? 

Reviewer #1: Yes

4. Have the authors made all data underlying the findings in their manuscript fully available?

Reviewer #1: Yes

5. Is the manuscript presented in an intelligible fashion and written in standard English?

Reviewer #1: Yes

6. Review Comments to the Author

Reviewer #1: The revised manuscript is improved and many of the comments have been incorporated.

I have few minor comments for authors to re-consider:

1. Line 84-86: “Patients with minimal pleural effusion which could not be tapped, history of bleeding diathesis or other contraindications to pleural fluid tapping and patients not willing to participate in the study were excluded.” This should be part of study participants and not study design. This was mentioned in previous review, but has been missed by the authors

2. The abstract should be edited and prepared in line with the main text of the manuscript.

3. The NPV has been changed in the Table, but not in the main text description or the abstract. Please review.

4. Line 90: It will be good to know the reference study/guideline used for the hypothesized sensitivity and specificity of GeneXpert for calculating Sample size.

7. PLOS authors have the option to publish the peer review history of their article (what does this mean?). If published, this will include your full peer review and any attached files.

Reviewer #1: No

---

## [Author Response · Author response to Decision Letter 2]

28 Feb 2021

15th of February 2021

The authors would like to thank the academic editor and reviewers for their specific and helpful comments. 

Please find enclosed the edited manuscript in word format with track changes.

Title: Diagnostic accuracy of GeneXpert MTB/RIF assay for detection of tubercular pleural

Effusion.

Author: Jyoti* Jain, Pooja$ Jadhao, Shashank^ Sharadchandra Banait, Preetam# Salunkhe.

Manuscript no: PONE-D-19-33113R2

The manuscript has been improved according to the suggestions:

REVIEWER #1: MINOR COMMENTS

REVIEWER COMMENT 1: Line 84-86: “Patients with minimal pleural effusion which could not be tapped, history of bleeding diathesis or other contraindications to pleural fluid tapping and patients not willing to participate in the study were excluded.” This should be part of study participants and not study design. This was mentioned in previous review, but has been missed by the authors.

AUTHOR RESPONSE: We are extremely sorry for missing this part. Now it has been changed as per your suggestions. Page 5, Line 81-88.

REVIEWER COMMENT 2: The abstract should be edited and prepared in line with the main text of the manuscript.

AUTHOR RESPONSE: The abstract has been changed as per your suggestions and now it is in line with the main text of the manuscript. Page 2, Line 28 and 29. The number of study population is mentioned in the abstract results.

REVIEWER COMMENT 3: The NPV has been changed in the Table, but not in the main text description or the abstract. Please review.

AUTHOR RESPONSE: We have corrected the error as suggested by you in not in the main text description and the abstract. Page 11, Line 186,187.

REVIEWER COMMENT 4: Line 90: It will be good to know the reference study/guideline used for the hypothesized sensitivity and specificity of GeneXpert for calculating Sample size.

AUTHOR RESPONSE: Reference and reference lists has been updated. Although the study quoted shown sensitivity of GeneXpert using bronchial washing or bronchoalveolar lavage fluid for the diagnosis of PTB was 81.6%, and specificity was 100%.We have hypothesized sensitivity of GeneXpert MTB/RIF assay for pleural TB = 0.60, hypothesized specificity of GeneXpert MTB/RIF assay for pleural TB = 0.99 for present study. 

Page 5, Line 96.

We hope that our modifications render our manuscript in its current form suitable for publication in PLOS ONE.

Yours sincerely,

---

## [Decision Letter · Decision Letter 3]

30 Apr 2021

Diagnostic accuracy of GeneXpert MTB/RIF assay for detection of tubercular pleural effusion

PONE-D-19-33113R3

Dear Dr. Jain,

We’re pleased to inform you that your manuscript has been judged scientifically suitable for publication and will be formally accepted for publication once it meets all outstanding technical requirements.

Kind regards,

Seyed Ehtesham Hasnain, Ph.D

Academic Editor

PLOS ONE

Additional Editor Comments (optional):

I have gone through the comments of both the reviewers, Authors response to the comments of the earlier Reviewers and also the re-revised manuscript. In my view, the Authors have comprehensively addressed all the issues raised by the Reviewers and brought a lot of clarity to the questions raised by the reviewers in the revised manuscript. Authors have uploaded the supporting information for figure S1 to address the comments of the reviewers. Authors have also changed the Table title numbering and the in-text Citations as per the reviewers suggestion. The manuscript is now recommended for publication given the extensive revision carried out.

Reviewers' comments:

Reviewer's Responses to Questions

**Comments to the Author**

1. If the authors have adequately addressed your comments raised in a previous round of review and you feel that this manuscript is now acceptable for publication, you may indicate that here to bypass the “Comments to the Author” section, enter your conflict of interest statement in the “Confidential to Editor” section, and submit your "Accept" recommendation.

Reviewer #3: All comments have been addressed

Reviewer #4: All comments have been addressed

2. Is the manuscript technically sound, and do the data support the conclusions?

Reviewer #3: Yes

Reviewer #4: Yes

3. Has the statistical analysis been performed appropriately and rigorously? 

Reviewer #3: Yes

Reviewer #4: Yes

4. Have the authors made all data underlying the findings in their manuscript fully available?

Reviewer #3: Yes

Reviewer #4: Yes

5. Is the manuscript presented in an intelligible fashion and written in standard English?

Reviewer #3: Yes

Reviewer #4: Yes

6. Review Comments to the Author

Reviewer #3: This study has corroborated similar results as evidenced in WHO policy update on use of XPERT MTB/RIF FOR THE DIAGNOSIS OF PULMONARY AND EXTRAPULMONARY TB, metaanalysis published in 2013 by WHO. Moreover, this study advocated genexpert assay can be used as rule out test for diagnosing TPN in resource limited settings. However, multicentric studies are further requested to update any new useful findings.

Reviewer #4: Dear Sir,

Current manuscript defined very nicely the utility of GeneXpert with its sensitivity and specificity. Authors have already justified the previous comments. Therefore, no need for further justifications.

7. PLOS authors have the option to publish the peer review history of their article (what does this mean?). If published, this will include your full peer review and any attached files.

Reviewer #3: No

Reviewer #4: No

---

## [Editor Report · Acceptance letter]

11 May 2021

PONE-D-19-33113R3 

Diagnostic accuracy of GeneXpert MTB/RIF assay for detection of tubercular pleural effusion 

Dear Dr. Jain:

I'm pleased to inform you that your manuscript has been deemed suitable for publication in PLOS ONE. Congratulations! Your manuscript is now with our production department. 

Kind regards, 

on behalf of

Prof. Seyed Ehtesham Hasnain 

Academic Editor

PLOS ONE